# NAM and TPB Approach to Consumers’ Decision-Making Framework in the Context of Indoor Smart Farm Restaurants

**DOI:** 10.3390/ijerph192114604

**Published:** 2022-11-07

**Authors:** Kyuhyeon Joo, Junghoon (Jay) Lee, Jinsoo Hwang

**Affiliations:** 1The College of Hospitality and Tourism Management, Sejong University, Seoul 143747, Korea; 2School of Hospitality Leadership, East Carolina University, 306 Rivers Building, Greenville, NC 27858-4353, USA

**Keywords:** indoor smart farm restaurants, the norm activation model, the theory of planned behavior, age

## Abstract

The movement toward smart farming, which has productivity and eco-friendly roles, is emerging in the foodservice industry in the form of indoor smart farm restaurants. The purpose of this study was to investigate the consumer decision-making processes in the context of indoor smart farm restaurants. The investigational framework was designed around the norm activation model (NAM) and the theory of planned behavior (TPB), with the moderating role of age. In particular, this study merged NAM and TPB to assess the effect of awareness of consumption consequences on consumers’ attitudes as well as the role played by subjective norms in the formation of personal norms. Data were collected from 304 respondents in South Korea. As a result of structural equation modeling, the proposed hypotheses of causal relationships were generally supported, excluding only the relationship between subjective norm and behavioral intention. The moderating role of age was identified in the relationships between (1) subjective norm and attitude, and (2) personal norm and behavioral intention. This study presents not only theoretical contributions as the first empirical study on consumer behavior in the context of indoor smart farm restaurants but also presents practical suggestions from the perspective of green marketing.

## 1. Introduction

It is widely known that conventional agriculture contributes to environmental pollution due to the use of pesticides and excessive carbon emissions [1,2]. In recent days, conventional agricultural production has also experienced increased risks due to the rise in fertilizer prices caused by the Russia-Ukraine conflict [3,4]. Several researchers have suggested a paradigm shift to the use of smart farms to make the agriculture sector more environmentally friendly and sustainable because smart farms are regarded as agricultural innovations that can overcome the issues mentioned above [5,6,7].

The smart agriculture market size is predicted to reach USD $29.23 billion by 2027 and will be able to sustainably provide food ingredients to consumers [8,9]. This trend also extends to the foodservice industry with enterprises like indoor smart farm restaurants (ISFR) that operate smart farms inside the store. For instance, the U.S. burger brand Good Stuff Eatery launched its first ISFR in South Korea [10], incorporating smart farms that allow consumers to see vegetables for their hamburgers and salads growing along the store’s walls. In Germany, the restaurant Good Bank, which opened in 2017, is a pioneer of the smart farm-to-table concept in the foodservice industry [11]. Likewise, the Radisson Blu Hotel set up smart farms inside Fish Market, its seafood restaurant in Dubai [12].

Despite this trend in the industry, empirical research on ISFRs is relatively lacking, especially from the perspective of consumer perceptions of and behavioral intentions toward sustainable food production/consumption. Consumers’ environmental concerns have increased during the COVID-19 pandemic, partly due to environmental pollution caused by disposable packaging [13,14]. That is, consumers are aware of the consequences of consumption, and they consider pro-environmental behavior in making consumption choices [15]. The norm activation theory (NAM) suggested by Schwartz [16] explains such a pro-social decision-making process. According to this theory, consumer awareness of consequences sequentially influences ascribed responsibility, personal norms, and pro-environmental behavior [16]. Since NAM is regarded as a robust theoretical model for predicting consumers’ pro-social behavior, previous studies have applied it in green research [17,18,19,20,21]. For instance, Nguyen [20] identified consumers’ intentions and behavior toward bringing their own shopping bags for environmental protection using the NAM framework. Kim et al. [21] successfully investigated consumers’ pro-environmental behavior in reducing food waste by applying the NAM framework. On the other hand, the theory of planned behavior (TPB) proposed by Ajzen [22] is an undoubtedly crucial theoretical model for predicting human decision-making. This theory comprises three concepts for predicting behavioral intentions: attitude, subjective norms, and perceived behavioral control. It also has been generally adopted to predict consumers’ pro-environmental behavior [23,24,25,26,27]. For example, Kumar et al. [26] predicted consumers’ pro-environmental purchase intentions to-wards eco-friendly apparel using the TPB framework. Kim et al. [27] also demonstrated consumers’ intentions to purchase home meal replacements in the context of eco-friendly TV home shopping broadcasts using the TPB framework. Whereas the NAM evokes pro-social motives, the TPB addresses self-interest motives that lead to human behavior [28]. Since these two theories explain different predictors for human behavior, each theoretical model can complement the motives explained by the other. Thus, this study proposes to merge these two theories to more comprehensively predict pro-environmental behavior.

Previous studies have also emphasized the crucial role of age in decision-making related to environmentally friendly behavior [29,30,31]. Roberts [30] found that older consumers have more ecological concerns and are more likely than younger consumers to make green purchases. Moon [29] also proved that the effect of perceived behavioral control and intention to purchase eco-friendly organic foods is moderated by age, with the older consumer group showing a higher path coefficient. On the other hand, since younger consumers tend to search for more information when making consumption choices [32], they are likely to give a greater degree of consideration to environmentally friendly behavior than older consumers. For instance, Wu et al. [31] found that the impact of tourists’ norms on pro-environmental behaviors is moderated by age and concluded that the younger tourist group presented a higher path coefficient. Since contradictions like these have arisen among studies related to the moderating role of age, this effort to investigate the moderating role of age in the context of ISFR is expected to provide meaningful contributions, both practically and academically.

In summary, this study employs a comprehensive research framework that integrates NAM and TPB with the moderating role of age. More specifically, the objectives of this paper are: (1) to apply TPB and NAM to predict consumers’ behavioral intentions in the context of ISFR; (2) to combine these two theoretical models; (3) to deepen the conceptual model by investigating the moderating role of age.

## 2. Literature Review and Hypotheses Development

### 2.1. Indoor Smart Farm Restaurants (ISFR)

Smart farms have higher labor efficiency and productivity than conventional agriculture as it uses automated agriculture based on an IoT system [33,34]. It is also a more environmentally friendly agriculture system than conventional agriculture, which contributes to environmental pollution through pesticides and carbon emissions [1,2]. Numerous professionals in academia suggest that a shift away from conventional agriculture toward smart farming will make the agriculture sector more environmentally friendly and sustainable [5,35,36].

The movement toward smart farming is also emerging in the foodservice industry in the form of the ISFR. Although foodservice is the largest energy user in the hospitality sector and emits around 490 tons of carbon dioxide produced per year per store [37,38], the integration of ISFR may minimize these environmental impacts. ISFR, which is exemplary in terms of green management due to its eco-friendly roles, can evoke consumers’ pro-social motives from the perspective of altruistic behavior. In addition, consumers’ self-interested motives are regarded as crucial predictors of behavioral intentions regarding restaurant choice [39,40,41]. It is necessary to consider both the altruistic and self-interested motives of consumers from the perspective of eco-friendly restaurants. In this sense, a comprehensive approach considering both motives related to ISFRs is meaningful in predicting consumer behavior.

### 2.2. Norm Activation Model (NAM)

Norm activation is defined as “the process in which people construct self-expectations regarding pro-social behavior” [42] (p. 323). The NAM suggested by Schwartz [16] in the altruistic context theorizes that an individual’s pro-social behavior is evoked by moral norms. This theoretical model consists of three concepts for predicting behavioral intention. They are awareness of consequences, ascription of responsibility, and personal norms [16]. It posits a sequential causal relationship in which awareness of consequences fosters the ascription of responsibility, strengthening personal norms and ultimately influencing behavioral intentions. First, awareness of consequences occurs when “someone is aware of the negative consequences for others or for other things one values when not acting prosaically” [43] (p. 426). It is the initial predictor of NAM because humans tend to feel responsibility dictated by the norms when they perceive negative outcomes of their behavior toward others [16]. Second, the ascription of responsibility manifests as “feelings of responsibility for the negative consequences of not acting pro-socially” [43] (p. 725). For instance, people tend to engage in pro-environmental behavior when they believe that the responsibility for environmental pollution is ascribed to themselves [44]. Third, personal norms relate to the feeling that it is a person’s “moral obligation to perform or refrain from specific actions” [45] (p. 191). Personal norm is regarded as the key construct within the norm activation process because it is the most proximal variable of behavioral intention in the theoretical model [16,46]. Some researchers have endorsed the moderation model of NAM, which argues that the causal relationship between personal norms and behavioral intentions is attenuated by awareness of consequences and ascribed responsibility [47,48]. Nevertheless, numerous studies also have provided strong evidence that NAM is the sequential mediator model [46,49]. In five separate articles, De Groot and Steg [42] compared interpretations of the moderation model and the mediator model, all of which verified the adequacy of NAM as the mediator model. Onwezen et al. [49] also supported the sequential norm activation process from the perspective of individual self-consciousness.

NAM is regarded as a robust theoretical model for predicting consumers’ pro-social purchasing behavior. It has been widely applied in the field of green research [17,19,21,31,50]. For instance, Shin et al. [19] applied NAM in the restaurant context to predict consumers’ choice of organic menu items. They showed that consumer awareness of environmental deterioration problems caused by the restaurant industry affected their ascription of responsibility. Ascription of responsibility also led them to consider their norms in choosing an eco-friendly organic menu when eating out, which led to behavioral intentions. Han et al. [50] also supported the NAM sequential mediator model in the context of green restaurants. Wang et al. [51] investigated tourists’ waste reduction behavioral intentions at tourist destinations using the NAM. Kim et al. [21] used this framework to predict consumers’ pro-environmental behavior in reducing food waste. Wu et al. [31] successfully investigated the environmentally responsible behavior of Chinese tourists using the NAM sequential mediator model. Govaerts and Olsen [17] used it in their study focused on seaweed consumption from the perspective of eco-friendly sustainable food sources. They stated that consumers should eat more seaweed to reduce the impact of food on the climate. This ascription of responsibility is evoked by an awareness of environmental consequences, and it activates personal norms of eating seaweed, which in turn facilitates behavioral intentions. Consequently, the norm activation model is considered a significant theoretical model in the context of environmentally friendly restaurants and sustainable food consumption. Given that the ISFR is a type of green restaurant using a sustainable agriculture system, it can be inferred that the ascription of responsibility caused by awareness of consequences would foster personal norms of using ISFR, thus leading to behavioral intentions. Based on the discussion above, the present study employed the sequential mediator model of NAM and proposed the following three hypotheses:

**Hypothesis** **1** **(H1).***Awareness of consequences influences ascription of responsibility*;

**Hypothesis** **2** **(H2).***Ascription of responsibility influences personal norms*;

**Hypothesis** **3** **(H3).***Personal norms influence behavioral intentions*.

### 2.3. Theory of Planned Behavior (TPB)

The TPB developed by Ajzen [22] extended from the theory of reasoned action (TRA), which states that an individual’s intention to behave or not behave ultimately influences the action [52]. This theory was regarded as the most influential theoretical model explaining humans’ volitional behavior [53,54]. The TRA comprises two concepts for predicting behavioral intentions. The first construct is an attitude, defined as “the degree to which a person has a favorable or unfavorable evaluation or appraisal of the behavior in question” [22] (p. 188). According to the TRA, an individual’s positive or negative attitude toward a particular behavior determines behavioral intentions [52,55]. The second construct is subjective norm, which refers to “the perceived social pressure to perform or not to perform the behavior” [22] (p. 188). It means that those who endorse a particular behavior are more likely to behave that way [52] because attitude and subjective norms positively influence behavioral intentions and, subsequently, volitional behavior as a result [52,53]. Ajzen [22] extended this theory by adding the concept of perceived behavioral control to further explain non-volitional behavior. Perceived behavioral control is “the perceived ease or difficulty of performing the behavior” [22] (p. 188). If an individual perceives their limitations in performing a particular behavior, they are less likely to form a behavioral intention in that direction [22,56]. After a meta-analytic review of both TPB and TRA, Armitage and Conner [57] stated that TPB has a higher efficacy as a predictor of behavior than TRA.

TPB has been widely adopted to predict consumers’ pro-environmental behavior [23,24,27,29,58]. Ching-Yu et al. [58] used TPB in the context of green restaurants and found that all three constructs (attitude, subjective norm, and perceived behavioral control) influence behavioral intention. Carfora et al. [23] adopted TPB to predict intentions to purchase eco-friendly organic milk. Moon [29] also proved that attitude, subjective norm, and perceived behavioral control positively affect the intention to purchase eco-friendly organic foods. Kim et al. [27] also successfully investigated consumers’ intentions to purchase home meal replacements in the context of eco-friendly TV home shopping broadcasts using the TPB framework. Dupont et al. [24] also applied TPB in a study focused on cultured meat consumption from the perspective of eco-friendly sustainable food sources. These two studies also proved the positive effect of all three predictors on behavioral intentions. TPB is an undoubtedly strong theoretical model for predicting consumers’ pro-environmental behavior. For example, a positive attitude toward visiting an ISFR would promote the behavioral intention to do so. If others endorse visiting an ISFR from the perspective of green consumption, consumers are more likely to volitionally behave positively towards visiting an ISFR. When consumers are sufficiently capable of visiting an ISFR, they would have positive behavioral intentions to do so. Based on the discussion above, the present study proposed the following three hypotheses:

**Hypothesis** **4** **(H4).***Attitude influences behavioral intentions*;

**Hypothesis** **5** **(H5).***Subjective norms influence behavioral intentions*;

**Hypothesis** **6** **(H6).***Perceived behavioral control influences behavioral intentions*.

In addition, since an individual’s attitude is explained as a learned predisposition, it can be fostered by others’ evaluations of a performed behavior [52,59], thus demonstrating that subjective norms can influence attitudes. Empirical studies based on TPB also found that subjective norms directly influence attitudes [60,61,62]. For instance, Tarkiainen and Sundqvist [62] found that subjective norms about buying organic products significantly affect attitudes. In the context of the hotel industry, Han et al. [61] used TPB and found that subjective norms concerning using green hotels cause consumers to develop favorable attitudes. Choe et al. [60] applied TPB in the context of an environmentally friendly edible insect restaurant, also proving that subjective norms influence attitudes. Accordingly, it can be inferred that subjective norms about using eco-friendly ISFR drive favorable attitudes. On this basis, the present study proposed the following hypothesis.

**Hypothesis** **7** **(H7).***Subjective norms influence attitude*.

### 2.4. Integrated Theoretical Models

NAM and TPB explain different predictors for consumer behavior. NAM explains pro-social motives, and TPB explains self-interest motives that foster human behavior [16,22,28]. Although these two models are undoubtedly convincing theories, neither model can fully capture predictors of behavioral intentions. To more comprehensively predict pro-environmental behavior, previous studies have complemented the framework by extending or integrating the theoretical model [63,64,65]. For instance, Han [63] used NAM to investigate an individual’s decision-making about environmentally responsible convention attendance but extended the study by adding attitude, anticipated feelings of pride and guilt, and social norms. The study found that awareness of consequences has a positive effect on attitudes, which in turn positively affects behavioral intentions. Le and Nguyen [64] also proved the causal relationship between awareness of consequences and attitude toward organic food purchases. Since awareness of consequences reflects an individual’s belief regarding the environment, it leads consumers to positive attitudes toward eco-friendly products/services [66,67,68].

In addition, previous studies have proposed that subjective norms precede personal norms [65,69,70]. Since subjective norms justify a particular behavior in society, they can be internalized as personal norms [3,71]. For instance, Han and Hyun [69] found the positive effect of subjective norms on personal norms in the context of an environmentally responsible museum. Similarly, Kim and Hwang [70] proved the causal relationship between subjective norms and personal norms in the context of an eco-friendly drone food delivery service. Choe et al. [39] also found that subjective norms affect personal norms in the context of an environmentally friendly edible insect restaurant. Based on the discussion above, it can be assumed that the awareness of consequences affects attitudes, whereas subjective norms affect personal norms in the context of ISFR. For the present study, the following two hypotheses integrating NAM and TPB in the context of ISFR were established:

**Hypothesis** **8** **(H8).***Awareness of consequences influences attitudes*;

**Hypothesis** **9** **(H9).***Subjective norms influence personal norms*.

### 2.5. Moderating Role of Age

In the field of consumer behavior, age is regarded as a crucial demographic factor in explaining pro-environmental behavior [30,72]. Previous studies have presented various points of view about the influencing role of age [32,73,74]. Roberts [30] argued that older consumers have more ecological concerns and a greater tendency to make green purchasing decisions than younger consumers. Similarly, Vining and Ebreo [73] stated that older consumers tend to exhibit more pro-environmental behavior than younger consumers. Hwang and Kim [75] also identified the moderating role of age in the context of eco-friendly drone food delivery services. The study demonstrated that age moderates the relationship between green image and attitude, with the older consumer group showing a higher path coefficient. Moon [29] also found a moderating role played by age in the TPB framework. That study proved that the effect of perceived behavioral control and intentions to purchase eco-friendly organic foods is moderated by age; the older consumer group showed a higher path coefficient.

Contrary to arguments that older consumers have a higher tendency toward pro-environmental behaviors than younger consumers, some research also argues that younger consumers are more likely to consider environmentally friendly behavior because they are inclined to search for more information and become engaged in social issues [32]. For example, Zimmer et al. [74] stated that younger consumers are more concerned with environmental issues than older consumers. Han et al. [76] also proved that younger consumers show higher behavioral intentions than older consumers in the context of green hotels. Likewise, Wu et al. [31] found a moderating role played by age in the NAM framework. That study demonstrated that the impact of tourists’ personal norms on pro-environmental behavior is moderated by age, indicating that the younger tourist group presents a higher path coefficient than the older tourist group. Regardless of these contradictory results about the moderating effect of age, research commonly supports the significant role age plays from the perspective of pro-environmental behavior, providing clear evidence for the moderating effect of age in the NAM and TPB framework. Thus, the present study proposed these hypotheses:

**Hypothesis** **10a** **(H10a).***The relationship between awareness of consequences and ascription of responsibility is moderated by age*;

**Hypothesis** **10b** **(H10b).***The relationship between ascription of responsibility and personal norms is moderated by age*;

**Hypothesis** **10c** **(H10c).***The relationship between personal norms and behavioral intentions is moderated by age*;

**Hypothesis** **10d** **(H10d).***The relationship between attitudes and behavioral intentions is moderated by age*;

**Hypothesis** **10e** **(H10e).***The relationship between subjective norms and behavioral intentions is moderated by age*;

**Hypothesis** **10f** **(H10f).***The relationship between perceived behavioral control and behavioral intentions is moderated by age*;

**Hypothesis** **10g** **(H10g).***The relationship between subjective norms and attitudes is moderated by age*;

**Hypothesis** **10h** **(H10h).***The relationship between awareness of consequences and attitudes is moderated by age*;

**Hypothesis** **10i** **(H10i).***The relationship between subjective norms and personal norms is moderated by age*.

### 2.6. Proposed Conceptual Model

The present study developed the conceptual model in Figure 1 based on the hypotheses proposed.

## 3. Methodology

### 3.1. Measurement Items

The present study employed measurement items developed from previous studies. The three predictors of behavioral intentions in the NAM (i.e., awareness of consequences, the ascription of responsibility, and personal norms) were measured using three measurement items each, drawn from Schwartz [16], Han et al. [50], and Govaerts and Olsen [17]. The three predictors of behavioral intentions in the TPB (i.e., attitudes, subjective norms, and perceived behavioral control) and behavior intentions were measured using three measurement items drawn from Ajzen [22], Ching-Yu et al. [58], and Choe et al. [39]. All seven constructs of the comprehensive framework integrating NAM and TPB were measured by 21 measurement items, and these items used a seven-point based Likert scale (1 = strongly disagree and 7 = strongly agree).

### 3.2. Data Collection

The current study collected data from the largest survey company with more than 1.5 million panelists in South Korea. The company sent the e-mail survey to 5792 panelists who had dined out within the last six months to identify respondents’ understanding of eating out. Before starting the survey, the respondents were given the video and article that fully explained ISFR and its environmentally friendly role. At the end of the survey, a gift of about US $1 was given to the respondents as a token of gratitude. Consequently, the company collected 330 samples, and this study used 304 data after eliminating 26 multivariate outliers.

## 4. Data Analysis

### 4.1. Profile of the Respondents

The respondent’s profile is shown in Table 1 (*n* = 304). Among the respondents, 48.0% were males (*n* = 146) and 52.0% were females (*n* = 158). The average age of the respondents was 37.0 years. The respondents with a monthly household income between USD $2001 and USD $3000, accounted for 28.6% (*n* = 87). The majority of the respondents were single (53.0%, *n* = 161) and had a bachelor’s degree (62.2%, *n* = 189).

### 4.2. Confirmatory Factor Analysis

This study conducted a confirmatory factor analysis to assess the measurement model. Table 2 presented measurement items, standardized factor loading values, average variance extracted value, and composite reliability value. All standardized loadings were higher than 0.7 and significant at *p* < 0.001, all constructs’ average variance extracted values were over 0.5, and composite reliability values were over 0.7 [77].

Table 3 presents mean and standard deviation values, correlation values, average variance extracted values, and the model fit of the measurement model (χ^2^(168) = 326.589, *p* < 0.001; χ^2^/df = 1.944, IFI = 0.975, CFI = 0.975, TLI = 0.969, and RMSEA = 0.056), and it had a satisfactory fit to the data. The squared correlation values were lower than the average variance extracted values for each construct [77].

### 4.3. Structural Equation Modeling

This study employed the structural equation modeling analysis to test the hypotheses. The structural model fit was presented in Table 4 (χ^2^(237) = 609.820, *p* < 0.001, χ^2^/df = 2.573, IFI = 0.948 CFI = 0.947, TLI = 0.939, and RMSEA = 0.071). Table 4 showed that eight paths were statistically significant except for H5. More specifically, awareness of consequences positively affects ascription of responsibility (*β* = 0.786 and *t* = 16.424). Awareness of consequences (*β* = 0.234 and *t* = 4.241) and subjective norms (*β* = 0.413 and *t* = 7.215) positively affect attitudes. Ascription of responsibility (*β* = 0.382 and *t* = 8.027) and subjective norms (*β* = 0.602 and *t* = 11.946) had a positive effect on personal norms. Lastly, personal norms (*β* = 0.333 and *t* = 5.726), attitudes (*β* = 0.352 and *t* = 6.938), and perceived behavior control (*β* = 0.523 and *t* = 6.938) positively affect behavioral intentions. Thus, H1, H2, H3, H4, H6, H7, H8, and H9 were supported.

### 4.4. Nested Model Comparisons

First, respondents (*n* = 304) were divided into two groups based on average age: (1) 148 respondents from the low age group, and (2) 156 respondents from the high age group. Table 5 presents the result of nested model comparisons in measurement and structural model. The test was verified prior to performing multiple-group analysis. Its comparative fit index difference (ΔCFI) between unconstrained and measurement weights is under 0.01 [78].

### 4.5. Moderating Role of Age

To test the moderating role of age, a multiple-group analysis was performed. The chi-square difference between the unconstrained and constrained models was used to prove the moderating role of age in each path of the integrated framework. The result of the analysis revealed that the effect of personal norms on behavioral intentions (Δχ^2^(1) = 4.179 > 3.84) and the effect of subjective norms on attitude (Δχ^2^(1) = 4.750 > 3.84) were moderated by age. More specifically, both paths show that the path coefficient of the high age group is higher than the low age group. Thus, H10c and H10g were supported. However, the other hypotheses were not statistically supported because their Δχ^2^(1) was lower than 3.84. The results of the multiple-group analysis are presented in Table 6, and the statistical results of this study are summarized in Figure 2.

## 5. Discussions and Implications

The present paper contributes theoretical extensions as the first study on consumer behavior in the context of ISFR. The study not only successfully applied a comprehensive framework combining two scientific approaches but also deepened the framework by investigating a moderating variable. Moreover, the study presents practical suggestions such as green advertisements for fostering awareness of consequences and influencer marketing strategies to enhance subjective norms. The theoretical and managerial implications of this study in detail are as follows.

### 5.1. Theoretical Implications

This study investigated the consumer’s decision-making framework in the context of ISFR using the NAM. This study focused on the pro-social motive fostered by smart farms’ eco-friendly benefits, such as low carbon emissions and no pesticides [1,2]. The current study adopted the mediator model of the NAM grounded on the literature (e.g., [16,46,49]). The study results supported three hypotheses of the sequential causal relationship: awareness of consequences → ascription of responsibility → personal norm → behavioral intentions. The results align with previous studies on green consumption [17,19,50]. They are theoretically valuable in explaining, for the first time, the process by which ISFR consumers form behavioral intentions in an altruistic context.

The study also suggested a comprehensive framework in the ISFR context based on integrating NAM with TPB. Since Ajzen [22] proposed TPB, the theory has been widely adopted to predict consumers’ pro-environmental behavior [23,24,58]. While TPB explains that self-interest motives evoke human behavior, NAM reveals pro-social motives [16,22,28]. Since each theoretical model can complement the motives of the other, the current study suggested integrating NAM and TPB for a more comprehensive prediction of pro-environmental behavior. This study also suggested two hypotheses related to (1) the effect of awareness of consequences on attitudes, and (2) the effect of subjective norms on personal norms [64,69,70]. The results revealed that all hypotheses were supported except for the effect of subjective norms on behavioral intentions (H5). Although this finding is somewhat different from findings in previous studies [18,22,70], there is sufficient evidence supporting it. For instance, Qi and Ploger [79] explained that subjective norms have an unstable predictive power in forming behavioral intentions, especially in the extended TPB in the context of green food purchases. Teixeira et al. [80] also reported insignificant consequences of the causal relationship between subjective norms and behavioral intentions in the context of organic food consumption. Despite the insignificant causal relationship between subjective norms and behavioral intentions, subjective norms are important predictors of attitudes and personal norms. Consequently, the current study theoretically implies a successful integration of NAM and TPB in the context of ISFR.

Finally, this study has deepened the comprehensive framework that integrates NAM and TPB by also investigating the moderating role of age. Age is a crucial demographic factor in explaining pro-environmental behavior [30,72]. Previous studies have presented contradictory points of view about age’s influence. Some studies concluded that older consumers tend to engage in pro-environmental behavior more than younger consumers because they have more ecological concerns [30,73]. Other studies suggested that younger, rather than older, consumers are more likely to consider environmentally friendly behavior because they tend to search for more information and are more in tune with social issues [32,74]. Age is widely adopted as a moderator in the pro-environmental behavior context, and previous studies using either NAM or TPB have also proven its moderating role [29,31,75]. Thus, this study proposed hypotheses related to the moderating role of age. The results revealed that age plays a moderating role in the effect of personal norms on behavioral intentions and the effect of subjective norms on attitude. In both instances, the path coefficient of the older age group is higher than that of the younger age group. These findings are in line with the findings of studies by Vining and Ebreo [73], Roberts [30], and Moon [29]. This study makes a first-time theoretical contribution in proving the moderating role of age in the context of ISFR.

### 5.2. Managerial Implications

The use of green advertisements will be a crucial strategy for the successful commercialization of ISFR, as they raise consumers’ awareness of consequences. The findings of this study suggest that ISFR marketers should plan green advertisements that stress not only the positive eco-friendly roles of smart farms but also the negative environmental consequences of the traditional agriculture and foodservice industry. Consumers may not consider the environmental pollution caused by the foodservice industry when they dine out. They also may not perceive the negative environmental influence of traditional agricultural products. Therefore, marketers who enhance consumers’ awareness of these negative influences also may be able to activate consumers’ personal norms. Since norm activation leads to constructing self-expectations regarding pro-social behavior [38], consumers who are aware of environmental consequences would be most likely to make environmentally friendly choices.

Study results also revealed that subjective norms drive personal norms and favorable attitudes, leading to behavioral intentions to use ISFR. Since subjective norms represent types of social perceptions of a particular behavior [22], promotional strategies also should emphasize ways of enhancing subjective norms. For instance, marketers can plan promotional content with social media influencers who endorse using ISFR to protect the environment. In addition, marketers could plan incentive promotions to offer coupons to consumers who promote the content of eco-friendly ISFR influencers. Doing so may extend endorsement of using ISFR to consumers’ friends and family or other important people.

The effect of subjective norms on attitudes and the effect of personal norms on behavioral intentions were both found to be moderated by age, with results for both paths showing a higher coefficient in the older consumer group. Therefore, overall marketing strategies for ISFRs should target older consumers for the most efficient performance. For instance, because social media can provide marketers with opportunities to target their consumers [81], ISFR marketers can direct advertisements or promotions to older consumers on social networks. They also can conduct focus group interviews with target-age consumers to help them establish or refine marketing strategies. These efforts would improve marketing efficiency for the successful commercialization of ISFR.

## 6. Conclusions

In this study of consumers’ behavioral intentions to use ISFR, a comprehensive framework integrating NAM and TPB was proposed while also investigating the moderating role of age. Data analysis revealed that all causal relationships were statistically supported, except for the effect of subjective norms on behavioral intentions, hypotheses 1, 2, 3, 4, 6, 7, 8, and 9. The analysis also found that age moderates the (1) effect of subjective norms on attitudes, and (2) the effect of personal norms on behavioral intentions, hypotheses 10c and 10g, were statistically supported. The two relationships show that the path coefficient of the high age group is higher than the low age group. The findings of this study present theoretical implications for and make practical contributions to the commercialization of ISFR. The study has explained for the first time the process by which consumers’ behavioral intentions to use ISFR could be formed in an altruistic context. In addition, it successfully investigated a comprehensive framework that integrates NAM and TPB in the context of ISFR. The study also found that age has a significant moderating effect in the suggested framework. Moreover, it has yielded practical suggestions for the commercialization of ISFR. Marketers of ISFR should evoke consumers’ awareness of consequences by planning green advertisements. They also can heighten consumers’ subjective norms through social media promotion with influencers. Ultimately, this study suggests that overall marketing strategies targeting older consumers would lead to efficient performance.

Nevertheless, the study does have some limitations. First, the 304 samples used in this study did not represent actual restaurant visitors because ISFR has not yet been fully commercialized in the market. Instead, ISFR and its eco-friendly role were explained to participants via articles and videos. To improve the generalizability of the findings, future studies may focus on qualitative research or test regressions using samples representing people who visited ISFRs and experienced their services and products. Second, the findings are difficult to generalize because this study collected only respondents from South Korea. It is widely known that food culture differs from region to region [82,83]. Thus, it is also significant to perform comparative research on consumers in areas where ISFRs are activated and areas where they are not. This study also limited its focus to pro-social and self-interested motives despite the rising popularity of new technology-based food services. Choe et al. [60] studied drone food delivery services from the technology acceptance perspective. Hwang et al. [84] also investigated the perceived risks of robotic restaurants and their influence. Therefore, investigating the technology acceptance perspective or perceived risks in the context of ISFR would offer meaningful insights for future studies. Finally, because customers’ word-of-mouth intentions and willingness to pay more are also crucial outcome variables in the pro-environmental behavior context [85,86], future research should examine these variables, especially how these may be predicted from the perspective of customers’ or operators’ pro-environmental behavior.

## Figures and Tables

**Figure 1 ijerph-19-14604-f001:**
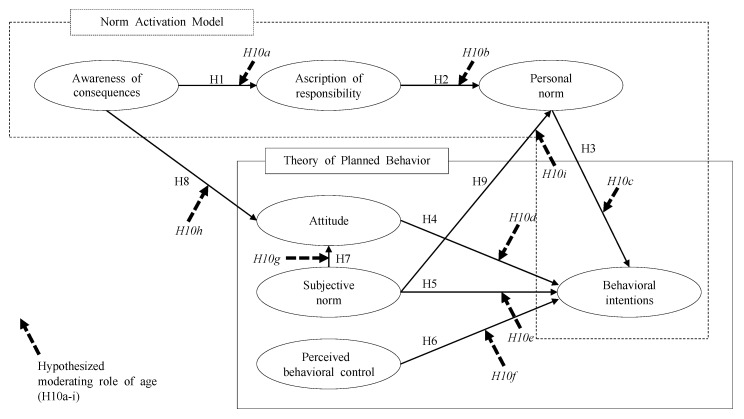
Proposed conceptual model. Note: H = hypothesis.

**Figure 2 ijerph-19-14604-f002:**
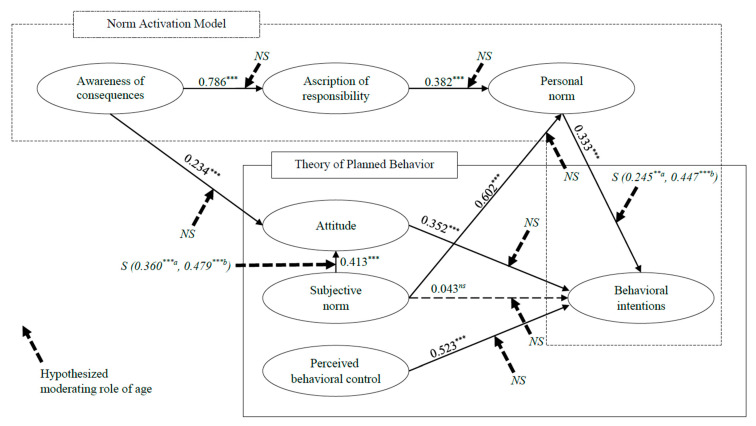
Standardized theoretical path coefficients. Notes 1: ** *p* < 0.01, *** *p* < 0.001, and *^ns^* not supported. Notes. 2: *S* = supported and *NS* = not supported. Notes 3: *^a^* path coefficient for a low age group and *^b^* path coefficient for a high age group.

**Table 1 ijerph-19-14604-t001:** Profile of respondents (*n* = 304).

Variables	*n*	%
**Gender**		
Male	146	48.0
Female	158	52.0
**Age (Mean = 37.02)**		
20s	88	28.9
30s	93	30.6
40s	92	30.3
50s	31	10.2
**Monthly income**		
Under $2000	54	17.8
$2001–3000	87	28.6
$3001–4000	66	21.7
$4001–5000	39	12.8
Over $5001	58	19.1
**Marital status**		
Single	161	53.0
Married	132	43.4
Widowed/Divorced	11	3.6
**Education level**		
Less than high school diploma	33	10.8
Associate degree	44	14.5
Bachelor’s degree	189	62.2
Graduate degree	38	12.5

**Table 2 ijerph-19-14604-t002:** Confirmatory factor analysis: items and loadings.

Construct and Scale Items	Standardized Loading *^a^*	AVE	CR
**Awareness of consequence**			
The foodservice industry can lead to environmental pollution (e.g., carbon emissions, food waste, disposable products).	0.897	0.852	0.945
The foodservice industry can potentially have a negative impact on global warming	0.951		
The foodservice industry can lead to the exhaustion of natural resources.	0.920		
**Ascription of responsibility**			
I believe that every restaurant customer is partly responsible for the environmental contaminants (e.g., carbon emission, food waste, disposable products, etc.) caused by the foodservice industry.	0.929	0.864	0.950
I feel that every restaurant customer is jointly responsible for the environmental deterioration caused by the environmental contaminants generated in the foodservice industry.	0.937		
Every restaurant customer must take partial responsibility for the environmental problems caused by the environmental contaminants generated in the foodservice industry.	0.923		
**Personal norm**			
I feel an obligation to choose an environmentally friendly way, such as ISFR when dining out.	0.904	0.778	0.913
Regardless of what other people do, because of my own values/principles I feel that I should behave in an environmentally friendly way when dining out.	0.843		
I feel it is important that consumers behave in a sustainable way, such as ISFR when dining out.	0.898		
**Attitude towards ISFR**			
Unfavorable–Favorable	0.820	0.807	0.926
Bad–Good	0.917		
Negative–Positive	0.952		
**Subjective norm**			
Most people who are important to me would think I should visit eco-friendly ISFR when I dine out.	0.911	0.883	0.958
Most people who are important to me would want to visit eco-friendly ISFR when I dine out.	0.955		
Most people who are important to me would prefer I visit eco-friendly ISFR when I dine out.	0.953		
**Perceived behavior control**			
Whether or not I visit eco-friendly ISFR when I dine out is completely up to me.	0.757	0.670	0.858
I’m confident that if I want, I can visit eco-friendly ISFR when I dine out.	0.914		
I have resources, time, and opportunities to visit eco-friendly ISFR when I dine out.	0.776		
**Behavior intentions**			
I will visit eco-friendly ISFR when I dine out.	0.891	0.932	0.821
I’m willing to visit eco-friendly ISFR when I dine out.	0.930		
I’m likely to visit eco-friendly ISFR when I dine out.	0.897		

Notes 1: *^a^* All factors loadings are significant at *p* < 0.001. Notes 2: AVE = Average variance extracted and CR = Composite reliabilities.

**Table 3 ijerph-19-14604-t003:** Descriptive statistics and associated measures.

Constructs	Items	Mean (SD)	(1)	(2)	(3)	(4)	(5)	(6)	(7)
(1) Awareness of consequence	3	5.77 (1.08)	**0.852 *^a^***	0.786 *^b^*	0.387	0.302	0.181	0.470	0.456
(2) Ascription of responsibility	3	5.47 (1.07)	0.618 *^c^*	**0.864**	0.506	0.255	0.268	0.383	0.431
(3) Personal norm	3	4.57 (1.27)	0.150	0.256	**0.778**	0.377	0.662	0.350	0.593
(4) Attitude	3	5.75 (1.19)	0.091	0.065	0.142	**0.807**	0.447	0.351	0.591
(5) Subjective norm	3	4.45 (1.29)	0.033	0.072	0.438	0.200	**0.883**	0.418	0.554
(6) Perceived behavior control	3	5.04 (1.25)	0.221	0.147	0.123	0.123	0.175	**0.670**	0.669
(7) Behavior intentions	3	5.26 (1.09)	0.208	0.186	0.352	0.349	0.307	0.448	**0.932**

Goodness-of-fit statistics: χ^2^(168) = 326.589, *p* < 0.001, χ^2^/df = 1.944, IFI = 0.975, CFI = 0.975, TLI = 0.969, and RMSEA = 0.056. Notes 1: SD = Standard deviation, IFI = Incremental fit index, CFI = Comparative fit index, TLI = Tucker-lewis index, and RMSEA = Root mean square error of approximation. Notes 2: *^a^* average variance extracted are along the diagonal, *^b^* correlations are above the diagonal, and *^c^* squared correlations are below the diagonal.

**Table 4 ijerph-19-14604-t004:** Standardized parameter estimates for the structural model.

	Path			Coefficients	*t*-Value	Hypothesis
H1	Awareness of consequence	→	Ascription of responsibility	0.786	16.424 ***	Supported
H2	Ascription of responsibility	→	Personal norm	0.382	8.027 ***	Supported
H3	Personal norm	→	Behavior intentions	0.333	5.726 ***	Supported
H4	Attitude	→	Behavior intentions	0.352	6.938 ***	Supported
H5	Subjective norm	→	Behavior intentions	0.043	0.748 *^ns^*	Not supported
H6	Perceived behavior control	→	Behavior intentions	0.523	9.738 ***	Supported
H7	Subjective norm	→	Attitude	0.413	7.215 ***	Supported
H8	Awareness of consequence	→	Attitude	0.234	4.241 ***	Supported
H9	Subjective norm	→	Personal norm	0.602	11.946 ***	Supported

Goodness-of-fit statistics: χ^2^(237) = 609.820, *p* < 0.001, χ^2^/df = 2.573, IFI = 0.948, CFI = 0.947, TLI = 0.939, and RMSEA = 0.071. Notes 1: *** *p* < 0.001 and *^ns^* not significant. Notes 2: IFI = Incremental fit index, CFI = Comparative fit index, TLI = Tucker-lewis index, and RMSEA = Root mean square error of approximation.

**Table 5 ijerph-19-14604-t005:** Result of measurement invariance: Nested model comparisons.

Measurement Model	χ^2^	df	IFI	TLI	CFI	RMSEA	ΔCFI
Unconstrained	541.122	336	0.969	0.960	0.968	0.045	
Measurement weights	548.187	350	0.970	0.963	0.969	0.043	0.001
Structural covariances	590.955	378	0.967	0.963	0.967	0.043	0.002
Measurement residuals	630.386	399	0.964	0.962	0.964	0.044	0.003
**Structural Model**	**χ^2^**	**df**	**IFI**	**TLI**	**CFI**	**RMSEA**	**ΔCFI**
Unconstrained	686.077	360	0.950	0.941	0.949	0.055	
Measurement weights	693.994	374	0.510	0.944	0.950	0.053	0.001
Structural weights	709.364	383	0.950	0.944	0.949	0.053	0.001
Structural covariances	711.122	386	0.950	0.945	0.949	0.053	0.000
Measurement residuals	761.777	411	0.946	0.944	0.945	0.053	0.004

Note: IFI = Incremental fit index, TLI = Tucker-lewis index, CFI = Comparative fit index, and RMSEA = Root mean square error of approximation.

**Table 6 ijerph-19-14604-t006:** Result for the moderating role of age.

Path		Unconstrained Model				Constrained Model	Test of Moderator	
		Low Age Group		High Age Group				
		*β*	*t*-Value	*β*	*t*-Value	χ^2^(360) = 686.077	χ^2^ Differences	Hypothesis
H10a	AoC→AoR	0.767	10.925 ***	0.801	12.172 ***	χ^2^(361) = 687.609	Δχ^2^(1) > 1.532	Not supported
H10b	AoR→PN	0.390	5.667 ***	0.394	5.983 ***	χ^2^(361) = 687.258	Δχ^2^(1) > 1.181	Not supported
H10c	PN→BI	0.245	3.060 **	0.447	8.670 ***	χ^2^(361) = 690.256	Δχ^2^(1) < 4.179	Supported
H10d	AT→BI	0.362	4.950 ***	0.346	4.875 ***	χ^2^(361) = 687.719	Δχ^2^(1) > 1.642	Not supported
H10e	SN→BI	0.084	1.066 *^ns^*	0.020	0.238 *^ns^*	χ^2^(361) = 686.754	Δχ^2^(1) > 0.677	Not supported
H10f	PBC→BI	0.563	7.041 ***	0.485	6.766 ***	χ^2^(361) = 687.852	Δχ^2^(1) > 1.775	Not supported
H10g	SN→AT	0.360	4.318 ***	0.479	6.113 ***	χ^2^(361) = 690.827	Δχ^2^(1) < 4.750	Supported
H10h	AoC→AT	0.290	3.530 ***	0.172	2.332 **	χ^2^(361) = 686.328	Δχ^2^(1) > 0.251	Not supported
H10i	SN→PN	0.588	8.193 ***	0.615	8.670 **	χ^2^(361) = 686.240	Δχ^2^(1) > 0.163	Not supported

Notes 1: AoC = Awareness of consequences, AoR = Ascription of responsibility, PN = Personal norm, BI = Behavioral intentions, AT = Attitude, SN = Subject norm, and PBC = Perceived behavioral control. Notes 2: ** *p* < 0.01, *** *p* < 0.001, and *^ns^* insignificant.

## Data Availability

Not applicable.

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
