# Peer review of "NAM and TPB Approach to Consumers’ Decision-Making Framework in the Context of Indoor Smart Farm Restaurants"

_ijerph, 2022, doi:10.3390/ijerph192114604_

Round 1

Reviewer 1 Report

Dear authors, in your work you have studied an interesting problem, but, unfortunately, which is not important for the development of scientific knowledge. I will try to explain.

You have done a number of studies, you have tried to combine and analyze two scientific approaches. That is, from the point of view of the requirements for publication, the scientific component in the form of methodology and research methodology is available in your work. But try to answer the question: what is the scientific novelty and practical significance of your research?

Moreover, the conclusions you have made about the significant role of age in choosing the planned behavior of consumers of smart restaurants matter only for a very limited group of consumers - citizens living in countries with a small area, a high standard of living, a high level of development of digitalization and insufficient natural resources. However, this approach is distinguished by all the "green science."

But you did a study in South Korea - a country that has certain conditions. Your conclusions may be important for this territory, but you did not describe in the article the special conditions of your territory. Moreover, you believe that these same problems may worry people living in other countries. For example, African or Central Asian countries, Russia, Latin America.

I believe it would be correct to compare the approaches you propose with the research of scientists from completely different countries of the world who have different conditions.

Author Response

We sincerely appreciate the reviewers’ comments and suggestions on the previous version of this manuscript. We have thoroughly studied all of them and have revised the manuscript accordingly. This report summarizes our responses to all the comments, which are in red for your convenience.

___________________________________________________________________________

Reviewer 1:

Dear authors, in your work you have studied an interesting problem, but, unfortunately, which is not important for the development of scientific knowledge. I will try to explain.

Response: Thank you for this critical comment. Our responses to your comments are summarized in the following section:

1. You have done a number of studies, you have tried to combine and analyze two scientific approaches. That is, from the point of view of the requirements for publication, the scientific component in the form of methodology and research methodology is available in your work. But try to answer the question: what is the scientific novelty and practical significance of your research?

Response: We thank you for this constructive review comment. Our study is the first study on consumer behavior in the context of indoor smart farm restaurants (hereafter ISFR), and successfully applied a comprehensive framework combining two scientific approaches. The study also presents practical suggestions for the commercialization of ISFR. These theoretical and managerial contributions are presented in in the ‘Discussions and Implications’ and ‘Conclusion’ sections. Following the reviewer’s comment, we additionally emphasized the scientific novelty and practical significance of the study in the beginning of the ‘Discussions and Implications’ section as follows.

The present paper contributes theoretical extensions as the first study on consumer behavior in the context of ISFR. The study not only successfully applied a comprehensive framework combining two scientific approaches, but also deepened the framework by investigating a moderating variable. Moreover, the study also presents practical suggestions such as green advertisements for fostering awareness of consequences and influencer marketing strategies to enhance subjective norms. The theoretical and managerial implications of this study in detail are as follows.

2. Moreover, the conclusions you have made about the significant role of age in choosing the planned behavior of consumers of smart restaurants matter only for a very limited group of consumers - citizens living in countries with a small area, a high standard of living, a high level of development of digitalization and insufficient natural resources. However, this approach is distinguished by all the "green science." But you did a study in South Korea - a country that has certain conditions. Your conclusions may be important for this territory, but you did not describe in the article the special conditions of your territory. Moreover, you believe that these same problems may worry people living in other countries. For example, African or Central Asian countries, Russia, Latin America. I believe it would be correct to compare the approaches you propose with the research of scientists from completely different countries of the world who have different conditions.

Response: Thank you for insightful comments. We also agree with your opinion of the necessity of the study in different conditions. Following the reviewer’s comment, we added the limitation of the study in the ‘Conclusion’ section as follows.

Second, the findings are difficult to generalize because this study collected only respondents in South Korea. It is widely known that food culture differs from region to region [82,83]. Thus, it is also significant to perform comparative research on consumers in areas where ISFRs are activated and areas where they are not.

82. Prescott, J.; Bell, G. Cross-cultural determinants of food acceptability: Recent research on sensory perceptions and preferences. Trends in Food Science & Technology 1995, 6, 201-205.

83. Tu, V.P.; Valentin, D.; Husson, F.; Dacremont, C. Cultural differences in food description and preference: Contrasting Vietnamese and French panellists on soy yogurts. Food Quality and Preference 2010, 21, 602-610.

Thank you so much for your comments and the time you gave for the improvement of our manuscript.

Reviewer 2 Report

The article is written in an appropriate way, but I think that the first person should be avoided.

In abstract it is missing findings, theoretical and practical implications.

In the introduction, the structure of the paper and some relevant references are missing.

In the literature review is missing some papers of 2021 and 2022.

In the methodology, the presentation of variables and their calculation should be improved.

The presentation of the results should be improved. Some tables and figures are not explained.

Conclusions are supported by results.

This paper is interesting for the readership of this journal. 

Author Response

We sincerely appreciate the reviewers’ comments and suggestions on the previous version of this manuscript. We have thoroughly studied all of them and have revised the manuscript accordingly. This report summarizes our responses to all the comments, which are in red for your convenience.

___________________________________________________________________________

Reviewer 2:

1. The article is written in an appropriate way, but I think that the first person should be avoided.

Response: Thanks for your comment and for allowing us to address them in more detail. The sentences in the first person were revised.

For the present study, we the following two hypotheses integrating NAM and TPB in the context of ISFR were established.

In this study of consumers’ behavioral intentions to use ISFR, we a comprehensive framework integrating NAM and TPB was proposed, while also investigating the moderating role of age.

2. In abstract it is missing findings, theoretical and practical implications.

Response: Thank you for pointing out this. We briefly added theoretical and practical implications in the ‘Abstract’ section.

This study not only presents theoretical contributions as the first empirical study on consumer behavior in the context of indoor smart farm restaurants, but also presents practical suggestions from the perspective of green marketing.

3. In the introduction, the structure of the paper and some relevant references are missing.

Response: Thank you for this comment. As you suggested, we revised the ‘Introduction’ section. In particular, we followed the guidelines for authors provided by the journal IJERPH as follows: (1) Emphasize the background and importance of the research at the beginning of the introduction, (2) Defining the purpose and significance of the main theories (NAM & TPB) and hypotheses (moderating role of age), (3) current state of the research field should be reviewed (empirical studies and reports in 2020-2022) and key publications (TPB: Ajzen, Fishbein; NAM: Schwartz) cited, and (4) briefly mentioning the main aim of the work. Following the reviewer’s comment, we added relevant references of 2022 as follows.

[Added Contents and References]

Nguyen [20] identified consumers' intentions and behavior toward bringing their own shopping bags for environmental protection using the NAM framework. Kim et al. [21] successfully investigated consumers' pro-environmental behavior in reducing food waste by applying the NAM framework. …

(omit)

… Kumar et al. [26] predicted consumers' pro-environmental purchase intentions to-wards eco-friendly apparel using the TPB framework. Kim et al. [27] also demonstrat-ed consumers' intentions to purchase home meal replacements in the context of eco-friendly TV home shopping broadcasts using the TPB framework.

20. Nguyen, T.P.L. Intention and behavior toward bringing your own shopping bags in Vietnam: integrating theory of planned behavior and norm activation model. Journal of Social Marketing 2022, 12, 395-419.

21. Kim, W.; Che, C.; Jeong, C. Food Waste Reduction from Customers’ Plates: Applying the Norm Activation Model in South Korean Context. Land 2022, 11, 109.

26. Kumar, N.; Garg, P.; Singh, S. Pro-environmental purchase intention towards eco-friendly apparel: Augmenting the theory of planned behavior with perceived consumer effectiveness and environmental concern. Journal of Global Fashion Marketing 2022, 13, 134-150.

27. Kim, H.M.; Lee, I.H.; Joo, K.; Lee, J.; Hwang, J. Psychological Benefits of Purchasing Home Meal Replacement in the Context of Eco-Friendly TV Home Shopping Broadcast: The Moderating Role of Personal Norm. International Journal of Environmental Research and Public Health 2022, 19, 7759.

[Appendix. Instructions for Authors]

The introduction should briefly place the study in a broad context and highlight why it is important. It should define the purpose of the work and its significance, including specific hypotheses being tested. The current state of the research field should be reviewed carefully and key publications cited. Please highlight controversial and diverging hypotheses when necessary. Finally, briefly mention the main aim of the work and highlight the main conclusions. Keep the introduction comprehensible to scientists working outside the topic of the paper.

(https://www.mdpi.com/journal/ijerph/instructions)

4. In the literature review is missing some papers of 2021 and 2022.

Response: Thanks for your comment. Following the reviewer’s comment, we added some papers of 2021 and 2022 as follows.

Wang et al. [51] investigated tourists’ waste reduction behavioral intentions at tourist destinations using the NAM. Kim et al. [21] used this framework to predict consumers' pro-environmental behavior in reducing food waste. Wu et al. [31] successfully investigated the environmentally responsible behavior of Chinese tourists using the NAM sequential mediator model. …

(omit)

…Moon [29] also proved that attitude, subjective norm and perceived behavioral control positively affect intention to purchase eco-friendly organic foods. Kim et al. [27] also successfully investigated consumers' intentions to purchase home meal replacements in the context of eco-friendly TV home shopping broadcasts using the TPB framework.

21. Kim, W.; Che, C.; Jeong, C. Food Waste Reduction from Customers’ Plates: Applying the Norm Activation Model in South Korean Context. Land 2022, 11, 109.

27. Kim, H.M.; Lee, I.H.; Joo, K.; Lee, J.; Hwang, J. Psychological Benefits of Purchasing Home Meal Replacement in the Context of Eco-Friendly TV Home Shopping Broadcast: The Moderating Role of Personal Norm. International Journal of Environmental Research and Public Health 2022, 19, 7759.

29. Moon, S.J. Investigating beliefs, attitudes, and intentions regarding green restaurant patronage: An application of the extended theory of planned behavior with moderating effects of gender and age. International Journal of Hospitality Management 2021, 92, 102727.

31. Wu, J.; Wu, H.C.; Hsieh, C.M.; Ramkissoon, H. Face consciousness, personal norms, and environmentally responsible behavior of Chinese tourists: Evidence from a lake tourism site. Journal of Hospitality and Tourism Management 2022, 50, 148-158.

51. Wang, S.; Ji, C.; He, H.; Zhang, Z.; Zhang, L. Tourists’ waste reduction behavioral intentions at tourist destinations: An integrative research framework. Sustainable Production and Consumption 2021, 25, 540-550.

5. In the methodology, the presentation of variables and their calculation should be improved.

Response: Thank you for this comment. In the ‘Methodology’ section, presentations of all variables equaled the names of variables in the literature review section, and the number of measurement items was also correctly calculated: (1) Each variable was measured using three measurement items. (2) The three constructs of the NAM (i.e., awareness of consequences, ascription of responsibility, and personal norms) were measured using 9 items. (3) The four constructs of the TPB (i.e., attitudes, subjective norms, perceived behavioral control, and behavior intentions) were measured using 12 items. (4) All seven variables were measured using 21 items. Following the reviewer’s comment, we tried to clearly explain the presentation of variables as follows.

The three predictors of behavioral intentions in the NAM (i.e., awareness of consequences, ascription of responsibility, and personal norms) were measured using three measurement items each, drawn from Schwartz [16], Han et al. [50], and Govaerts and Olsen [17]. The three predictors of behavioral intentions in the TPB (i.e., attitudes, subjective norms, and perceived behavioral control) and behavior intentions were each measured using three measurement items, drawn from Ajzen [22], Ching-Yu et al. [58], and Choe et al. [39]. All seven constructs of the comprehensive framework integrating NAM and TPB were measured by 21 measurement items, and these items used a seven-point based Likert scale (1 = strongly disagree and 7 = strongly agree).

6. The presentation of the results should be improved. Some tables and figures are not explained.

Response: Thanks for your comment. All figures and tables had been cited and explained in the ‘Data analysis’ section. Following the reviewer’s comment, we supplemented explanations of some tables as follows.

Table 2 presented measurement items, standardized factor loading values, average variance extracted value, and composite reliability value. All standardized loadings were higher than .7 and significant at p < .001, all constructs’ average variance extracted values were over .5 and composite reliability values were over .7 [77].

Table 3 presented mean and standard deviation value, correlation values, average variance extracted values, and the model fit of measurement model (χ2(168) = 326.589, p < .001; χ2/df = 1.944, IFI = .975, CFI = .975, TLI = .969, and RMSEA = .056), and it had a satisfactory fit to the data. The squared correlation values were lower than the average variance extracted values for each construct [77].

Table 5 presented the result of nested model comparisons in measurement and structural model. The test was verified prior to performing multiple-group analysis, its comparative fit index difference (ΔCFI) between unconstrained and measurement weights is under .01 [78].

7. Conclusions are supported by results.

Response: Thank you for this comment. In the ‘Conclusion’ section, we summarized the results, implications, and limitations. Following the reviewer’s comment, we supplemented the contents of the results in more detail as follows.

Data analysis revealed that all causal relationships were statistically supported with the exception of the effect of subjective norms on behavioral intentions, hypothesis 1, 2, 3, 4, 6, 7, 8, and 9 were supported. The analysis also found that age moderates the (1) effect of subjective norms on attitudes and the (2) effect of personal norms on behavioral intentions, hypothesis 10c and 10g were statistically supported. The two relationships show that the path coefficient of the high age group is higher than the low age group.

8. This paper is interesting for the readership of this journal.

Response: Thank you for this encouragement and the time you gave for the improvement of our manuscript.

Reviewer 3 Report

Dear Authors,

Thank you for the opportunity to review your work. The idea of Indoor Smart Farm restaurant is both unique and relevant to the core of the journal.

Also, the idea development and its presentation is equally very well done. However, I have few observations that the authors may consider reviewing in order to improve the work even more.

1. Background of indoor smart farm restaurant is missing in the abstract, the abstract can be updated to include more detail about this concept so as to help the readers grasp the full idea from the abstract. Similarly, the abstract is devoid of the findings of the study and more importantly the implications of these findings.

2. Similar to the last comment above, it will be lovely if the authors can include an additional paragraph in the introduction dedicated to articulating the implication, contribution or significance of the study. more like your unique selling point.

3. starting a sentence with "because" is not an acceptable practice in English, the authors are advised to check through the manuscript and make neccessary corrections.

4.  the format of the presentation of the results in table 3 is confusing, please stick to the universal standard and report square root of AVE along the diagonal, keep correlations of items below the diagonal and put AVE, Composite reliability and factor loadings together in table 2.

Author Response

We sincerely appreciate the reviewers’ comments and suggestions on the previous version of this manuscript. We have thoroughly studied all of them and have revised the manuscript accordingly. This report summarizes our responses to all the comments, which are in red for your convenience.

___________________________________________________________________________

Reviewer 3:

Dear Authors,

Thank you for the opportunity to review your work. The idea of Indoor Smart Farm restaurant is both unique and relevant to the core of the journal. Also, the idea development and its presentation is equally very well done. However, I have few observations that the authors may consider reviewing in order to improve the work even more.

Response: Thank you for the kind suggestions. Thank you for this encouragement and for all of your comments. Our responses to your comments are summarized in the following section:

1. Background of indoor smart farm restaurant is missing in the abstract, the abstract can be updated to include more detail about this concept so as to help the readers grasp the full idea from the abstract. Similarly, the abstract is devoid of the findings of the study and more importantly the implications of these findings.

Response: We added the background of indoor smart farm restaurants in the “Abstract” section as follows.

The movement toward smart farming which has productivity and eco-friendly roles is emerging in the foodservice industry in the form of the indoor smart farm restaurants.

2. Similar to the last comment above, it will be lovely if the authors can include an additional paragraph in the introduction dedicated to articulating the implication, contribution or significance of the study. more like your unique selling point.

Response: Thank you for this helpful comment. We briefly added theoretical and practical implications in the ‘Abstract’ section.

This study not only presents theoretical contributions as the first empirical study on consumer behavior in the context of indoor smart farm restaurants, but also presents practical suggestions from the perspective of green marketing.

3. starting a sentence with "because" is not an acceptable practice in English, the authors are advised to check through the manuscript and make neccessary corrections.

Response: Thank you for this comment. Sentences starting with “because” were revised as follows.

Because smart farms are regarded as agricultural innovations that can overcome the issues mentioned above, a number of researchers have suggested a paradigm shift to the use of smart farms to make the agriculture sector more environmentally friendly and sustainable (e.g., [5,6,7]).

A number of researchers have suggested a paradigm shift to the use of smart farms to make the agriculture sector more environmentally friendly and sustainable because smart farms are regarded as agricultural innovations that can overcome the issues mentioned above (e.g., [5,6,7]).

Because Since contradictions like these have arisen among studies related to the moderating role of age, this effort to investigate the moderating role of age in the context of ISFR is expected provide meaningful contributions, both practically and academically.

Because a smart farm uses automated agriculture based on an IoT system, it has higher labor efficiency and productivity than conventional agriculture [29,30].

Smart farms have higher labor efficiency and productivity than conventional agriculture as it uses automated agriculture based on an IoT system [34,35].

Because Since subjective norms represent types of social perceptions of a particular behavior [22], promotional strategies also should emphasize ways of enhancing subjective norms.

4. the format of the presentation of the results in table 3 is confusing, please stick to the universal standard and report square root of AVE along the diagonal, keep correlations of items below the diagonal and put AVE, Composite reliability and factor loadings together in table 2.

Response: We thank you for this constructive review comment. We revised the Table 2 and 3 as you suggested.

Table 2. Confirmatory factor analysis: items and loadings.

Construct and scale items Standardized
Loadinga
AVE CR
Awareness of consequence      
The foodservice industry can lead to environmental pollution (e.g., carbon emissions, food wastes, disposable products). .897 .852 .945
The foodservice industry can potentially have a negative impact on global warming .951
The foodservice industry can lead to the exhaustion of natural resources.  .920
(omit)

Notes 1: a All factors loadings are significant at p < .001. Notes 2: AVE = Average variance extracted and CR = Composite reliabilities.

Table 3. Descriptive statistics and associated measures.

Constructs Items Mean (SD) (1) (2) (3) (4) (5) (6) (7)
(1) Awareness of consequence 3 5.77 (1.08) .852a .786b .387 .302 .181 .470 .456
(2) Ascription of responsibility 3 5.47 (1.07) .618c .864 .506 .255 .268 .383 .431
(omit)

Goodness-of-fit statistics: χ2(168) = 326.589, p < .001, χ2/df = 1.944, IFI = .975, CFI = .975, TLI = .969, and RMSEA = .056. Notes 1: SD = Standard deviation, IFI = Incremental fit index, CFI = Comparative fit index, TLI = Tucker-lewis index, and RMSEA = Root mean square error of approximation. Notes 2: a average variance extracted are along the diagonal, b correlations are above the diagonal, and c squared correlations are below the diagonal.

Thank you so much for your comments and the time you gave for the improvement of our manuscript.

_______________
